psychology

open data, open badges, reproducibility, open science, meta-research, journal policy

**Author for correspondence:**
Tom E. Hardwicke
e-mail: tom.hardwicke@uva.nl

# Analytic reproducibility in articles receiving open data badges at the journal *Psychological Science*: an observational study

Tom E. Hardwicke[1,2], Manuel Bohn[3], Kyle MacDonald[4], Emily Hembacher[5], Michèle B. Nuijten[6], Benjamin N. Peloquin[5], Benjamin E. deMayo[5], Bria Long[5], Erica J. Yoon[5] and Michael C. Frank[5]

[1]Department of Psychology, University of Amsterdam, Amsterdam, The Netherlands
[2]Meta-Research Innovation Center Berlin (METRIC-B), QUEST Center for Transforming Biomedical Research, Charité – Universitätsmedizin, Berlin, Germany
[3]Department of Comparative Cultural Psychology, Max Planck Institute for Evolutionary Anthropology, Leipzig, Germany
[4]Department of Communication, University of California, Los Angeles, CA, USA
[5]Department of Psychology, Stanford University, Stanford, CA, USA
[6]Department of Methodology and Statistics, Tilburg School of Social and Behavioral Sciences, Tilburg University, Tilburg, The Netherlands

TEH, 0000-0001-9485-4952; MB, 0000-0001-6006-1348;
KM, 0000-0001-6111-3824; EH, 0000-0001-6578-4238;
MBN, 0000-0002-1468-8585; BNP, 0000-0002-4876-9906;
BEdM, 0000-0001-8723-6419; BL, 0000-0001-7156-6878;
EJY, 0000-0002-7895-6390; MCF, 0000-0002-7551-4378

For any scientific report, repeating the original analyses upon the original data should yield the original outcomes. We evaluated analytic reproducibility in 25 *Psychological Science* articles awarded open data badges between 2014 and 2015. Initially, 16 (64%, 95% confidence interval [43,81]) articles contained at least one 'major numerical discrepancy' (>10% difference) prompting us to request input from original authors. Ultimately, target values were reproducible without author involvement for 9 (36% [20,59]) articles; reproducible with author involvement for 6 (24% [8,47]) articles; not fully reproducible with no substantive author response for 3 (12% [0,35]) articles; and not fully reproducible despite author involvement for 7 (28% [12,51]) articles. Overall, 37 major numerical discrepancies remained out of 789 checked values (5% [3,6]), but original conclusions did not appear affected.

Non-reproducibility was primarily caused by unclear reporting of analytic procedures. These results highlight that open data alone is not sufficient to ensure analytic reproducibility.

## 1. Introduction

A minimum quality standard expected of all scientific manuscripts is that any reported numerical values can be reproduced if the original analyses are repeated upon the original data [1]. This concept is known as *analytic reproducibility* ([2]; or relatedly, *computational reproducibility*,[1] [3]). When a number cannot be reproduced, this minimally indicates that the process by which it was calculated has not been sufficiently documented. Non-reproducibility may also indicate that an error has occurred, either during the original calculation or subsequent reporting. Either way, the integrity of the analysis pipeline that transforms raw data into reported results cannot be guaranteed. As a result, non-reproducibility can undermine data reuse [5], complicate replication attempts [7], and create uncertainty about the provenance and veracity of scientific evidence, potentially undermining the credibility of any associated inferences [2].

Difficulty establishing the analytic reproducibility of research reports has been encountered in several scientific domains, including economics, political science, behavioural ecology and psychology [3–6,8,9]; but see [10]. A preliminary obstacle for many such studies is that research data are typically unavailable [11–14]. Even when data can be accessed, suboptimal data management and inadequate documentation can complicate data reuse [5,15]. These failures to adequately preserve and share earlier stages of the research pipeline typically prevent downstream assessment of analytic reproducibility.

A previous study in the domain of psychology largely circumvented data availability issues by capitalizing on a mandatory data sharing policy introduced at the journal *Cognition* [5]. In a sample of 35 articles with available data that had already passed initial quality checks, 24 (69%) articles contained at least one value that could not be reproduced within a 10% margin of error. Reproducibility issues were resolved in 11 of the 24 articles after consultation with original authors yielded additional data or clarified analytic procedures. Ultimately, 64 of 1324 (5%) checked values could not be reproduced despite author involvement. In some cases, the data files contained errors or were missing values and at least one author published a correction note. Importantly, there were no clear indications that the reproducibility issues seriously undermined the conclusions of the original articles. Nevertheless, this study highlighted a number of quality control and research transparency issues including suboptimal data management; unclear, incomplete, or incorrect analysis specification; and reporting errors.

In the present study, we intended to investigate whether the findings of Hardwicke *et al.* [5] extended to a corpus of *Psychological Science* articles that received an 'open data badge' to signal data availability. In order to focus specifically on the reproducibility of the analysis process rather than upstream data availability or data management practices, we selected only articles that had reusable data according to a previous study [15]. The submission guidelines of *Psychological Science* specifically stated that authors could earn an open data badge for 'making publicly available the digitally shareable data necessary to reproduce the reported result'.[2] Additionally, authors were asked to self-certify that they had provided '…sufficient information for an independent researcher to reproduce the reported results'.[3] If this policy was operating as intended, all numbers presented in these articles should be independently reproducible. If not, we hoped to learn about causes of non-reproducibility in order to identify areas for improvement in journal policy and research practice. Thus, the aim of the study was to assess the extent to which data shared under the *Psychological Science* open badge scheme actually enabled analytic reproducibility.

## 2. Methods

The study protocol (hypotheses, methods and analysis plan) was pre-registered on 18 October 2017 (https://osf.io/2cnkq/). All deviations from this protocol or additional exploratory analyses are explicitly acknowledged in electronic supplementary material, section D.

---

[1]Computational reproducibility is often assessed by attempting to re-run original computational code and can therefore fail if original code is unavailable or non-functioning (e.g. [3,4]). By contrast, analytic reproducibility is assessed by attempting to repeat the original analysis procedures, which can involve implementing those procedures in new code if necessary (e.g. [5,6]).

[2]See https://perma.cc/SFV8-DAZ6 (originally retrieved 9 October 2017).

[3]See https://perma.cc/N8K7-DXP9?type=image (originally retrieved 9 October 2017).

## 2.1. Design

We employed a retrospective non-comparative case-study design based on Hardwicke et al. [5]. The numerical discrepancy between *original values* reported in the target articles and *reanalysis values* obtained in our reproducibility checks was quantified using percentage error[4] (PE = |reanalysis−original|/original × 100). Numerical discrepancies were classified as 'minor' (0% > PE < 10%) or 'major' (PE ≥ 10%). If an original *p*-value fell on the opposite side of the alpha boundary relative to a reanalysis *p*-value we additionally recorded a 'decision error'. The alpha boundary was assumed to be 0.05 unless otherwise stated. We recorded when there was insufficient information to complete aspects of the analyses.

We classified articles as 'not fully reproducible' (any major numerical discrepancies, decision errors, or insufficient information to proceed) or 'reproducible' (no numerical discrepancies or minor numerical discrepancies only) and noted whether author involvement was provided or not. We recorded the potential causal loci of non-reproducibility and judged the likelihood that non-reproducibility impacted conclusions drawn in the original articles. Team members provided a subjective estimate of the time they spent on each reproducibility check. Additional design details are available in electronic supplementary material, section A.

## 2.2. Sample

The sample was based on a corpus of psychology articles that had been examined in a previous project investigating the impact of an 'open badges' scheme introduced at *Psychological Science* [15]. This sample was selected because the open badges scheme and assessment of reusability by Kidwell et al. [15] enabled us to largely circumvent upstream issues related to data availability and data management and instead focus on downstream issues related to analytic reproducibility. A precision analysis indicates that the sample size affords adequate precision for the purposes of gauging policy compliance (electronic supplementary material, section B).

Of 47 articles marked with an open data badge, Kidwell and colleagues had identified 35 with datasets that met four reusability criteria (accessible, correct, complete and understandable). For each of these articles, one investigator (T.E.H.) attempted to identify a coherent set of descriptive and inferential statistics (e.g. means, standard deviations, *t*-values, *p*-values; figure 3), roughly 2–3 paragraphs of text, sometimes including a table or figure, related to a 'substantive' finding based on 'relatively straightforward' analyses. We focused on substantive findings because they are the most important and straightforward analyses to ensure that our team had sufficient expertise to re-run them. In total, 789 discrete numerical values reported in 25 articles published between January 2014 and May 2015 were designated as target values. Further information about the sample is available in electronic supplementary material, section B.

## 2.3. Procedure

The aim of the reproducibility checks was to recover the target original values by repeating the original analysis (as described in the original articles and any additional documentation such as supplementary materials, codebook, analysis scripts) upon the original data. Attempting alternative analyses was outside the scope of the study, even if the original analysis seemed suboptimal or erroneous.

To minimize error and facilitate reproducibility, each reproducibility check was conducted by at least two team members who documented the reanalysis process in an R Markdown report (available at https://osf.io/hf9jy/). If articles were initially classified as not fully reproducible, we emailed the author(s) of the article and used any additional information they provided to try and resolve the issues before a final classification was determined. Additional procedural details are available in electronic supplementary material, section A.

## 2.4. Data analysis

The results can be considered at several layers of granularity. Detailed qualitative information about each reproducibility check is available in the individual reproducibility reports (https://osf.io/hf9jy/) and

---

[4]An important caveat of this measure should be noted: large percentage differences can occur when the absolute magnitude of values is small (and vice versa). Thus, when considering the consequences of non-reproducibility for individual cases, quantitative measures should be evaluated in the full context of our qualitative observations (electronic supplementary material, section E).

**Table 1.** Final outcomes of reproducibility checks at the article level after original authors were contacted, *n* (%, [95% CI]).

| | target values fully reproducible? | |
| --- | --- | --- |
| | yes | no |
| **original author involvement?** | | |
| yes | 6 (24%, [8,47]) | 7 (28%, [12,51]) |
| no | 9 (36%, [20,59]) | 3 (12%, [0,35])[6] |

summarized in a short 'vignette' available in electronic supplementary material, section E. We report descriptive and inferential[5] statistics at the article level and value level. Ninety-five per cent confidence intervals (CIs) are displayed in square brackets.

The present study is highly comparable in goal and design to a previous study [5], creating an opportunity to synthesize their estimates of analytic reproducibility. To this end, we used random-effects models with inverse-variance weighting to meta-analyse the raw proportion estimates of article-level analytic reproducibility for the two studies, separately before and after original authors were contacted [16].

# 3. Results

Prior to seeking original author involvement, all target values in 9 out of the 25 articles (36%, CI [19,57]) were reproducible, with the remaining 16 articles (64%, CI [43,81]) containing at least one major numerical discrepancy. After requesting input from original authors, several issues were resolved through the provision of additional information and ultimately all target values in 15 (60%, CI [39,78]) articles were reproducible, with the remaining 10 (40%, CI [22,61]) articles containing at least one major numerical discrepancy. In all except one case, author input involved provision of additional information that was not included in the original article or clarification of original information that was ambiguous or incomplete. In the exception case (4-1-2015_PS), the authors pointed out that we had missed a relevant footnote in the original article. In three cases, the authors did not respond and in three cases there was insufficient information to proceed with the reanalysis. The full breakdown of reproducibility outcomes after author contact are shown in table 1 and the results of a meta-analysis synthesizing the findings with those of a previous study [5] are displayed in figure 1. In no cases did the observed numerical discrepancies appear to be consequential for the conclusions stated in the original articles (see electronic supplementary material, section E).

After author involvement, 37 major numerical discrepancies remained among the 789 target values examined across all articles (5%, CI [3,6]). This included two decision errors for which we obtained a statically significant *p*-value in contrast to a reported non-significant *p*-value and one decision error with the opposite pattern. A scatterplot illustrating the consistency of original and reanalysis *p*-values is displayed in figure 2. The frequency of numerical discrepancies by value types is displayed in figure 3.

Where possible we attempted to identify the causal locus of the reproducibility issues we encountered though this was not always possible to confirm definitively. Electronic supplementary material, figure S2 shows the frequency of four types of discrete causal loci that we determined were involved in non-reproducibility and how many of these issues were resolved through original author input. The most common issues we encountered were related to unclear, incomplete, or incorrect reporting of analytic procedures. Examples include unidentified statistical tests, unclear aggregation procedures, non-standard *p*-value reporting and unreported data exclusions. Most of these issues could be resolved when original authors provided additional information. Less frequently, we encountered typographical errors and some issues related to data files, including erroneous or missing data. For many instances of non-reproducibility, the causal locus remained unidentified even after contact with original authors.

---

[5]Note that although one goal of the study was to characterize analytic reproducibility within a finite sample as a follow-up to Kidwell *et al.* [15], we were also interested in generalizing to other psychology articles. To address this second goal, we use standard inferential statistics, recognizing that, because they are generated from a convenience sample, the generality of our findings may be limited (for more details, see Discussion).

[6]Note that although this confidence interval technically includes 0, we are confident that 0 is not a possible population proportion due to the existence of non-reproducible cases that we document here.

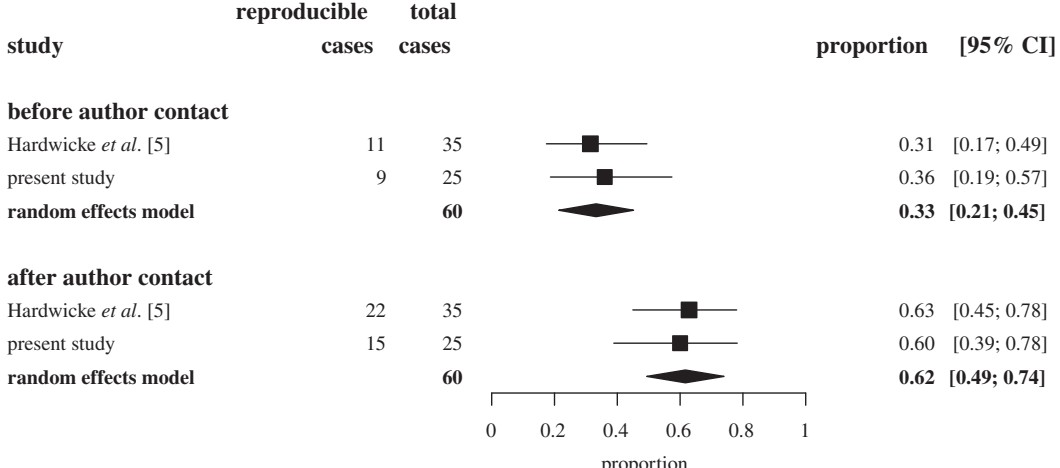

**Figure 1.** Forest plot showing the proportion of articles with fully reproducible target values in Hardwicke et al. [5] and the present study before and after contacting original authors for assistance with reproducibility checks. Squares represent individual study proportions. Diamonds represent summary proportions estimated by random effects models. Error bars represent 95% confidence intervals (CIs).

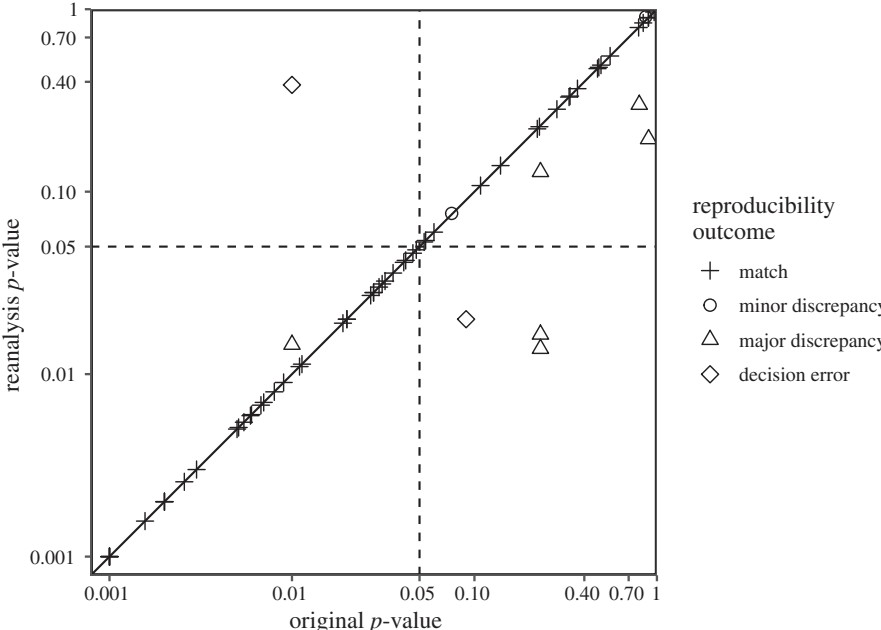

**Figure 2.** Scatterplot showing reanalysis p-values as a function of original p-values, classified by reproducibility outcome. Axes are on a log scale. The diagonal line represents perfect consistency between reanalysis and original values. Dashed lines represent the typical alpha threshold (0.05) for statistical significance. Original values reported relative to thresholds (e.g. $p < 0.05$) are represented by the threshold value when there was a major discrepancy and represented by the reanalysis value when there was a match. For display purposes, 41 values below 0.001 are not shown. Note that the two values in the bottom right quadrant of the graph have been conservatively designated as 'major errors' rather than 'decision errors' due to uncertainty about how the original analysis was performed. Specifically, insufficient information was provided in the original article (ID 9-5-2014_PS) about how multiplicity corrections were applied (see Vignette 25 in electronic supplementary material, section E for further information).

Team members provided concurrent estimates of their time spent on each stage of the analysis. Altogether, they estimated that they spent between 1 and 30 (median = 7, interquartile range = 5) hours actively working on each reproducibility check (electronic supplementary material, figure S3; total time = 213 h). This estimate excludes time spent waiting on internal (within our team) and external (with original authors) communications. Availability of original analysis scripts appeared to provide

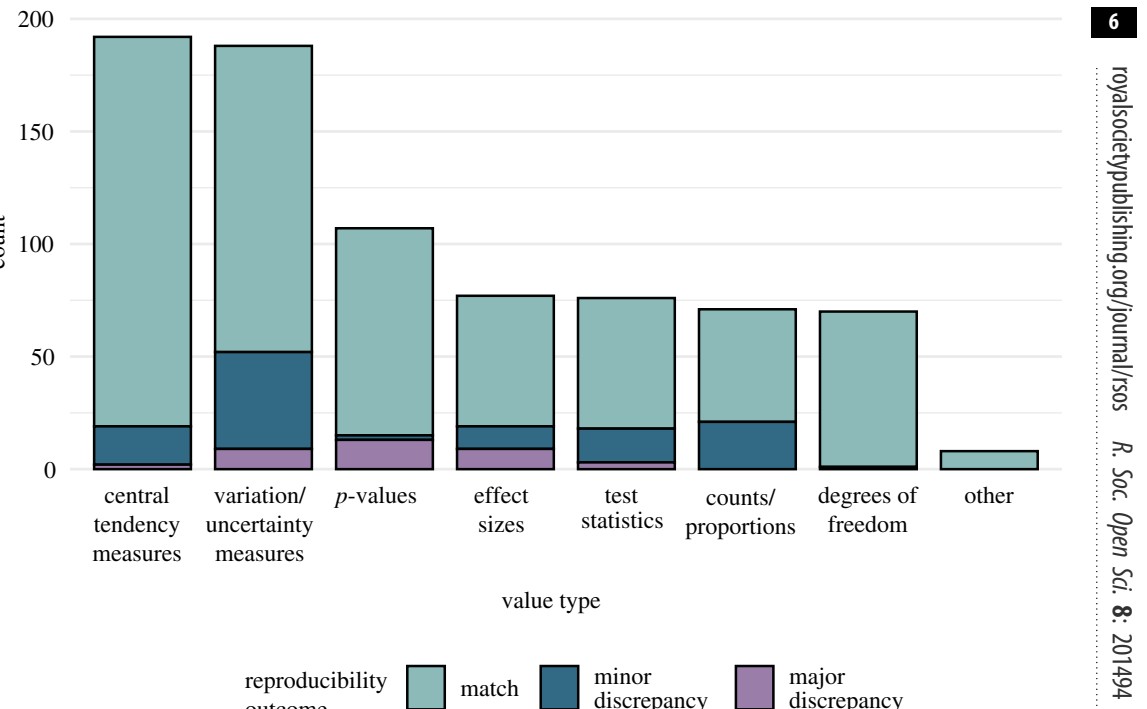

**Figure 3.** Frequency of reproducibility outcomes by value type. Variation/uncertainty measures include standard deviations, standard errors and confidence intervals. Effect sizes include Cohen's $d$, Pearson's $r$, partial eta squared and phi. Test statistics include $t$, $F$ and $\chi^2$. Central tendency measures include means and medians.

some modest benefits in terms of reproducibility outcomes and time expenditure (see electronic supplementary material, section C).

## 4. Discussion

A reasonable expectation of any scientific manuscript is that repeating the original analyses upon the original data will yield the same quantitative outcomes [1]. We have found that this standard of analytic reproducibility was frequently not met in a sample of *Psychological Science* articles receiving open data badges. Importantly, none of the reproducibility issues we encountered appeared seriously consequential for the conclusions stated in the original articles (though in three cases, insufficient information about original analytic procedures prevented us completing our reproducibility checks). Nevertheless, non-reproducibility highlighted fundamental quality control and documentation failures during data management, data analysis and reporting. Furthermore, the extensive time investment required to check and establish analytic reproducibility would be likely to be prohibitive for many researchers interested in building upon and re-using data, for example, to conduct robustness checks, novel analyses, or meta-analyses. The findings are consistent with a previous study which found a similar rate of non-reproducibility and identified unclear, incomplete, or incorrect analysis reporting as a primary contributing factor [5]. Although the open badges scheme introduced at *Psychological Science* has been associated with an increase in data availability [15], the current findings suggest that additional efforts may be required in order to ensure analytic reproducibility.

Non-reproducibility does not always imply that prior conclusions based on the original results are fatally undermined—indeed we encountered no such scenarios in this study. Determining the consequences of non-reproducibility for the interpretation of results is not always straightforward and is to some extent subjective. We considered several factors including the precision of original hypotheses, the extent and magnitude of non-reproducibility, and whether there were $p$-value decision errors. For example, in a case where a hypothesis only makes a directional prediction, a reanalysis indicating a statistically significant but smaller effect than originally reported, would still be consistent with the original hypothesis. Or in a case where effect sizes and $p$-values can be reproduced successfully, but one finds a few major numerical errors in the descriptive statistics, the most obvious cause is a reporting error, rather than an error in the implementation of the original analysis, and the

original conclusions are not obviously compromised by this. Therefore, scientific reasoning is needed to evaluate the consequences of non-reproducibility on a case-by-case basis.

Some reproducibility issues were resolved after input from original authors; however, relying on such assistance is not ideal for several reasons: (i) it substantially increases the workload and time investment, both for researchers attempting to re-use data and for original authors; (ii) information related to the study is more likely to be forgotten or misplaced over time, reducing the ability of authors to assist [6,17]; (iii) authors may be unwilling or unable to respond, as was the case for three of the articles we examined. Author provision of additional information or clarifications beyond what was previously reported highlighted that non-reproducibility was often caused by unclear, incomplete, or incorrect reporting of analytic procedures in published papers. In some cases, authors informed us of errors in shared data files or analysis scripts, suggesting that at some stage the original data, analyses, and research report had become decoupled. In some cases, neither we nor the original authors could reproduce the target values and the causal locus of non-reproducibility remained unidentified; it was no longer possible to reconstruct the original analysis pipeline.

This study has a number of important limitations and caveats. Most pertinently, we have reported confidence intervals and conducted a random-effects meta-analysis to aid generalization of the findings to a broader population; however, such generalizations should be made with caution as several characteristics of our sample are likely to have had a positive impact on the reproducibility rate. We specifically selected a sample of articles for which data were already available and screened against several reusability criteria [15]. By contrast, most psychology articles are unlikely to be accompanied by raw data or analysis scripts [11,13,14], and even when data is available, suboptimal management and documentation can complicate reuse [5]. Consequently, most psychology articles would be likely to fail a reproducibility check before reaching the stage of the analysis pipeline at which we began our assessments.

Additionally, all articles in the sample had been submitted to a leading psychology journal that had recently introduced a number of policy changes intended to improve research rigour and transparency [18]. These included the open badges scheme, removing word limits for methods and results sections, and requesting explicit disclosure of relevant methodological information like data exclusions. It is likely that these new initiatives positively impacted reproducibility rates, either through directly encouraging adoption of more rigorous research practices or attracting researchers to the journal who were already inclined to use such practices [19]. Some evidence, for example, implies that data availability is modestly associated with a lower prevalence of statistical reporting inconsistencies [20]. In sum, several features of the current sample are likely to facilitate reproducibility relative to a more representative population of psychology articles.

While the current findings are concerning, non-reproducibility is fortunately a solvable problem, at least in principle. Journals are well-situated to influence practices related to research transparency; for example, journal data sharing mandates have been associated with marked increases in data availability [5,21]. Further requirements to share analysis scripts may also enhance reproducibility as veridical documentation of the analysis process is often poorly captured in verbal prose [5]. However, as with data sharing, mere availability may not be sufficient to confer the potential benefits of analysis scripts. The utility of scripts is likely to depend on a range of factors, including clarity, structure and documentation, as well the programming language that is used and whether they are stored in a proprietary format or require special expertise to understand. In the present study, script availability appeared to offer modest benefits in terms of reproducibility outcomes and time expenditure; however, as only six articles shared scripts,[7] it is difficult to draw strong conclusions about their impact from this evidence alone. Journals could also conduct independent assessment of analytic reproducibility prior to publication, as has been adopted by the *American Journal of Political Science* [22]; however, this would naturally require additional resources (for discussion see [23]). Ideally, initiatives intended to improve analytic reproducibility should undergo empirical scrutiny in order to evaluate costs and benefits and identify any policy shortfalls or unintended consequences [12].

Although journal policy is potentially helpful for incentivisation and verification, the reproducibility of a scientific manuscript is fundamentally under the control of its authors. Fortunately, a variety of tools are now available that allow for the writing of entirely reproducible research reports in which data, analysis code and prose are intrinsically linked (e.g. [24]). It should be noted that ensuring the reproducibility of a scientific manuscript requires a non-trivial time investment and presently, there is

---

[7]Note that *Psychological Science*'s author guidelines explicitly state that the criteria for an open data badge includes sharing 'annotated copies of the code or syntax used for all exploratory and principal analyses'. See https://perma.cc/SFV8-DAZ6 (originally retrieved 9 October 2017).

room for improvement in the data management and analysis practices adopted by psychology researchers [25]. Continued development of user-friendly tools that facilitate reproducibility and dedicated reproducibility training may help to reduce this burden. Additionally, the costs of ensuring reproducibility could be offset by the benefits of improved workflow efficiency, error detection and mitigation, and opportunities for data reuse. Detailed guidance on data sharing, data management and analytic reproducibility is available to support psychological scientists seeking to improve these practices in their own research [26]. The present manuscript is an illustration of these practices in action (https://doi.org/10.24433/CO.1796004.v3).

## 5. Conclusion

Together with previous findings [5], this study has highlighted that the analytic reproducibility of published psychology articles cannot be guaranteed. It is inevitable that some scientific manuscripts contain errors and imperfections, as researchers are only human and people make mistakes [27]. However, most of the issues we encountered in this study were entirely avoidable. Data availability alone is insufficient; further action is required to ensure the analytic reproducibility of scientific manuscripts.

## 6. Open practices statement

The study protocol (hypotheses, methods and analysis plan) was pre-registered on the Open Science Framework on 18 October 2017 (https://osf.io/2cnkq/). All deviations from this protocol or additional exploratory analyses are explicitly acknowledged in electronic supplementary material, section D. We report how we determined our sample size, all data exclusions, all manipulations and all measures in the study. All data, materials and analysis scripts related to this study are publicly available on the Open Science Framework (https://osf.io/n3dej/). To facilitate reproducibility, this manuscript was written by interleaving regular prose and analysis code using knitr [28] and papaja [29], and is available in a Code Ocean container (https://doi.org/10.24433/CO.1796004.v3) which re-creates the software environment in which the original analyses were performed. Analysis code, reports and Code Ocean containers are also available for each reproducibility check (electronic supplementary material, section E).

Data accessibility. All data, materials and analysis scripts related to this study are publicly available on the Open Science Framework (https://osf.io/n3dej/). To facilitate reproducibility, this manuscript was written by interleaving regular prose and analysis code using knitr [28] and papaja [29], and is available in a Code Ocean container (https://doi.org/10.24433/CO.1796004.v3) which re-creates the software environment in which the original analyses were performed.

Authors' contributions. T.E.H. and M.C.F. designed the study. T.E.H., M.B., K.M., E.H., M.B.N., B.N.P., B.E.d.M., B.L., E.J.Y. and M.C.F. conducted the reproducibility checks. T.E.H. performed the data analysis. T.E.H. and M.C.F. wrote the manuscript. M.B. and M.B.N. provided feedback on the manuscript. All authors gave final approval for publication.

Competing interests. We declare we have no competing interests.

Funding. T.E.H.'s contribution was enabled by a general support grant awarded to the Meta-Research Innovation Center at Stanford (METRICS) from the Laura and John Arnold Foundation and a grant from the Einstein Foundation and Stiftung Charité awarded to the Meta-Research Innovation Center Berlin (METRIC-B).

Acknowledgements. We are grateful to the authors of the original articles for their assistance with the reproducibility checks. We thank students from Stanford's Psych 251 class, who contributed to the initial reproducibility checks.

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
