## [Reviewer comments · Royal Society Open Science]

Review History

RSOS-201494.R0 (Original submission)

Review form: Reviewer 1 (Gustav Nilsson)

Is the manuscript scientifically sound in its present form?

Yes

Are the interpretations and conclusions justified by the results?

Yes

Is the language acceptable?

Yes

Do you have any ethical concerns with this paper?

No

Have you any concerns about statistical analyses in this paper?

No

Recommendation?

Accept with minor revision (please list in comments)

Comments to the Author(s)

Please see attached file (Appendix A).

Review form: Reviewer 2 (Rink Hoekstra)**Is the manuscript scientifically sound in its present form?**

Yes

Are the interpretations and conclusions justified by the results?

Yes

Is the language acceptable?

Yes

Do you have any ethical concerns with this paper?

No

Have you any concerns about statistical analyses in this paper?

No

Recommendation?

Accept with minor revision (please list in comments)

Comments to the Author(s)

I believe the research question the authors try to answer is a very important one in the current academic climate, in which the need for transparent research is often propagated, but rarely studied. Using badges, whether you like them or not, is one way to encourage researchers to be more open, and it's very interesting to see whether badges actually improve the quality of papers. To be clear: I think transparency is pivotal for science to enable the essential self-correcting mechanism to function properly, and I don't think that more transparency necessarily creates higher quality research, but it would be a welcome side-effect.

So I was excited to read this paper, and it does not disappoint. It is well-written, clear, thorough, and it deals well with potential shortcomings. Despite the fact that I've told a friend yesterday (before having read this paper) that I usually write long reviews, I think this one can be very short: I think the authors did a nice and painstaking job (I don't envy having to check hundreds of outcomes), and the results are useful for the current debate around transparency. So I'd have no problem with endorsing this paper, barring a few (very) minor issues:

- alpha = .05 is taken as the criterion for decision errors. Did the authors check whether this was indeed the significance level the authors used? Usually it does, but some authors prefer other levels of alpha, and in that case labelling decision errors when they are on the other side of .05 might be premature
- Personally, I would have put the footnote at the end of page 7 in the text: I think this is quite important
- Minor detail: the crosses in Figure 2 seem like downward diagonal lines to me, since the upward lines coincide with the main upward diagonal (so at first I thought: where are the crosses?)
- Confidence intervals are weird, and confidence intervals for proportions close to 0 and 1 may be even weirder. Having a CI from $[0, X]$ makes sense given the correct interpretation of CIs

("in the long run,...."), but shouldn't be interpreted as if 0 were a plausible value (as a matter of fact, the authors have shown it's not 0). I have no particular suggestion how to deal with that (there is no mistake or anything), but it may be good to stress this point, possibly near Supplementary Figure 1

But as said before, I applaud the authors for writing this paper, and it was a pleasure reading it.

I always sign my reviews.
Rink Hoekstra

Decision letter (RSOS-201494.R0)

Dear Dr Hardwicke

On behalf of the Editors, we are pleased to inform you that your Manuscript RSOS-201494 "Analytic reproducibility in articles receiving open data badges at Psychological Science: An observational study" has been accepted for publication in Royal Society Open Science subject to minor revision in accordance with the referees' reports. Please find the referees' comments along with any feedback from the Editors below my signature.

Please submit your revised manuscript and required files (see below) no later than 7 days from today's (ie 05-Nov-2020) date. Note: the ScholarOne system will 'lock' if submission of the revision is attempted 7 or more days after the deadline. If you do not think you will be able to meet this deadline please contact the editorial office immediately.

Kind regards,

Anita Kristiansen
Editorial Coordinator

on behalf of Dr Denes Szucs (Associate Editor) and Essi Viding (Subject Editor)

Reviewer comments to Author:

Reviewer: 1

Comments to the Author(s)

Please see attached file

Reviewer: 2

Comments to the Author(s)

I believe the research question the authors try to answer is a very important one in the current academic climate, in which the need for transparent research is often propagated, but rarely studied. Using badges, whether you like them or not, is one way to encourage researchers to be more open, and it's very interesting to see whether badges actually improve the quality of papers. To be clear: I think transparency is pivotal for science to enable the essential self-correcting mechanism to function properly, and I don't think that more transparency necessarily creates higher quality research, but it would be a welcome side-effect.

So I was excited to read this paper, and it does not disappoint. It is well-written, clear, thorough, and it deals well with potential shortcomings. Despite the fact that I've told a friend yesterday (before having read this paper) that I usually write long reviews, I think this one can be very short: I think the authors did a nice and painstaking job (I don't envy having to check hundreds of outcomes), and the results are useful for the current debate around transparency. So I'd have no problem with endorsing this paper, barring a few (very) minor issues:

- alpha = .05 is taken as the criterion for decision errors. Did the authors check whether this was indeed the significance level the authors used? Usually it does, but some authors prefer other levels of alpha, and in that case labelling decision errors when they are on the other side of .05 might be premature
- Personally, I would have put the footnote at the end of page 7 in the text: I think this is quite important
- Minor detail: the crosses in Figure 2 seem like downward diagonal lines to me, since the upward lines coincide with the main upward diagonal (so at first I thought: where are the crosses?)
- Confidence intervals are weird, and confidence intervals for proportions close to 0 and 1 may be even weirder. Having a CI from $[0, X]$ makes sense given the correct interpretation of CIs ("in the long run,....."), but shouldn't be interpreted as if 0 were a plausible value (as a matter of fact, the authors have shown it's not 0). I have no particular suggestion how to deal with that (there is no mistake or anything), but it may be good to stress this point, possibly near Supplementary Figure 1

But as said before, I applaud the authors for writing this paper, and it was a pleasure reading it.

I always sign my reviews.

Rink Hoekstra

===PREPARING YOUR MANUSCRIPT===

===PREPARING YOUR REVISION IN SCHOLARONE===

-- Ensure that your data access statement meets the requirements at <https://royalsociety.org/journals/authors/author-guidelines/#data>. You should ensure that you cite the dataset in your reference list. If you have deposited data etc in the Dryad repository, please only include the 'For publication' link at this stage. You should remove the 'For review' link.

-- If you have uploaded ESM files, please ensure you follow the guidance at <https://royalsociety.org/journals/authors/author-guidelines/#supplementary-material> to include a suitable title and informative caption. An example of appropriate titling and captioning may be found at https://figshare.com/articles/Table_S2_from_Is_there_a_trade-off_between_peak_performance_and_performance_breadth_across_temperatures_for_aerobic_scops_in_teleost_fishes_/3843624.

Author's Response to Decision Letter for (RSOS-201494.R0)

See Appendix B.

Decision letter (RSOS-201494.R1)

Dear Dr Hardwicke,

It is a pleasure to accept your manuscript entitled "Analytic reproducibility in articles receiving open data badges at Psychological Science: An observational study" in its current form for publication in Royal Society Open Science. The comments of the reviewer(s) who reviewed your manuscript are included at the foot of this letter.

on behalf of Dr Denes Szucs (Associate Editor) and Essi Viding (Subject Editor)
openscience@royalsociety.org

Appendix A

Review of "Analytic reproducibility in articles receiving open data badges at Psychological Science: An observational study", Hardwicke et al.

2020-09-19

Signed: Gustav Nilsson, gustav.nilsson@ki.se

Summary of findings

This preregistered study evaluated analytic reproducibility in 25 articles in Psychological Science awarded open data badges 2014-2015. The team identified target findings in each article and tried to reproduce them using the available data. The authors report that target values were reproducible without original author involvement for 9 of the 25 articles; reproducible with original author involvement for 6 articles; not fully reproducible with no substantive author response for 3 articles; and not fully reproducible despite author involvement for 7 articles. Non-reproducibility was primarily caused by unclear reporting of analytic procedures. These results are similar to those reported in an earlier study (Hardwicke et al. 2018) of computational reproducibility of articles with open data in the journal Cognition.

Scope of this review

I have read and reviewed the manuscript including the supplements. I have read the preregistration and verified that the preregistration was followed, except for changes which are transparently reported, and which are clearly and well motivated. I have reviewed the main analysis code, which is clear and well-annotated. I have re-run the CodeOcean "reproducible run" that generates the main manuscript; I can confirm that it works and generates the manuscript, and that the main tables and plots are the same as in the submitted manuscript. I have spot-checked one of the reproducibility reports (id 1-1-2015-PS), in which I cursorily examined the completeness of the documentation of extraction of target values and the analysis process; as far as I can see it is very well documented.

Main assessment

The evaluation of computational reproducibility is an important metascientific endeavour. The present study adds important corroboration to estimates from the earlier study (Hardwicke et al. 2018) of computational reproducibility in psychology. Both this study and the previous one are limited by the sampling frame. For this reason, these new results are important, and their similarity to the previous results increases the likelihood that the results are representative of the field. The study is of high quality, particularly as regards the excellent methods documentation, including reproducible code.

My main comments concern the interpretation of the results. Presently, the discussion is rather abstract, and could have stood largely unchanged regardless what the results had been. I would invite the authors to reason explicitly about whether any other outcome could have led to a different conclusion. Personally, I think this paper contributes meaningfully towards estimating the field-wide computational reproducibility in psychology. The use of a random-effects meta-analysis implies that the authors are trying to do this, but the discussion only broaches this point indirectly, by pointing to limitations of generalizability.

There are some other points that could also be discussed, if the authors wish. One point I find intriguing is that conclusions in the target articles often seem to be unaffected by major numerical errors. Does this imply a gap between the theoretical model and the statistical test? Is there scope for improvement here?

Another point of interest concerns systemic factors for quality control. The authors rightly point to the role of journals in setting policy and in conducting quality checks. They also point out resources available to scientists for generating reproducible workflows. But there is considerable scope for further discussion of systemic factors. Why, for instance, do universities send out numbers with their name on it without having had a second person double-check the work of a typically junior person? What institutional responsibilities are there? Can we (university-affiliated scientists) learn from other types of research performing organizations?

Finally, I wonder what the authors think that we still need to find out about computational reproducibility. Are we done with psychology now and ready to move on to other fields? Is there a research agenda that would motivate digging deeper here? For instance, would it be interesting to study covariates associated with computational reproducibility? Should we do intervention studies to evaluate effects of new policies?

Minor points

I offer the following points at the authors' discretion.

- line 75 (about): I suggest to add here that in some cases data were missing or incorrect, and that the work led the authors of several papers to issue corrections.
- line 91: I recommend to state the aim clearly here at the end of the introduction (preferably verbatim as in the preregistration: “does open data shared under the Open Sciences Badges scheme actually enable successful analytic reproducibility”).
- table 1: I suggest to use a 2x2 table instead, for better readability.
- figure 1: I suggest to draw the x axis from 0 to 1, for better readability, or else to truncate it equally at the top and bottom.

- figure 2: Why are not all p-values in upper left and lower right quadrants decision errors? I must be missing something. Can the authors please elaborate on this to clarify.

- Suppl. A: The authors write that "The most serious consequence of non-reproducibility is that it undermines the credibility of associated scientific claims." While I am inclined to agree, I do not think that this statement is obviously true. I recommend that the authors give arguments supporting this statement.

- line 521 (about): It would be interesting to know which of the eligibility criteria were not met in the different cases.

Potential conflicts of interest

I am a member of the badges committee of the Center for Open Science, which defines the criteria for earning the open science badges, and I served for several years as its chair. I was a co-author of the earlier study (Hardwicke et al. 2018) which investigated computational reproducibility of articles in the journal *Cognition* in a manner similar to this study.

Appendix B

Response to reviewers

Manuscript number: RSOS-201494

Manuscript title: Analytic reproducibility in articles receiving open data badges at Psychological Science: An observational study

We are grateful to the editor and reviewers for their in-depth assessment of the manuscript and helpful comments. Below we provide a point-by-point response to each of the reviewers' comments and describe changes we have made to the manuscript.

RC: Reviewer comment

AR: Author response

Red text: Additional or modified text included in the manuscript

Reviewer #1

RC1:

Review of “Analytic reproducibility in articles receiving open data badges at Psychological Science: An observational study”, Hardwicke et al.
2020-09-19

Signed: Gustav Nilsson, gustav.nilsson@ki.se

Summary of findings

This preregistered study evaluated analytic reproducibility in 25 articles in Psychological Science awarded open data badges 2014-2015. The team identified target findings in each article and tried to reproduce them using the available data. The authors report that target values were reproducible without original author involvement for 9 of the 25 articles; reproducible with original author involvement for 6 articles; not fully reproducible with no substantive author response for 3 articles; and not fully reproducible despite author involvement for 7 articles. Non-reproducibility was primarily caused by unclear reporting of analytic procedures. These results are similar to those reported in an earlier study (Hardwicke et al. 2018) of computational reproducibility of articles with open data in the journal Cognition.

Scope of this review

I have read and reviewed the manuscript including the supplements. I have read the preregistration and verified that the preregistration was followed, except for changes which are transparently reported, and which are clearly and well motivated. I have reviewed the main

analysis code, which is clear and well-annotated. I have re-run the CodeOcean “reproducible run” that generates the main manuscript; I can confirm that it works and generates the manuscript, and that the main tables and plots are the same as in the submitted manuscript. I have spot-checked one of the reproducibility reports (id 1-1-2015-PS), in which I cursorily examined the completeness of the documentation of extraction of target values and the analysis process; as far as I can see it is very well documented.

Main assessment

The evaluation of computational reproducibility is an important metascientific endeavour. The present study adds important corroboration to estimates from the earlier study (Hardwicke et al. 2018) of computational reproducibility in psychology. Both this study and the previous one are limited by the sampling frame. For this reason, these new results are important, and their similarity to the previous results increases the likelihood that the results are representative of the field. The study is of high quality, particularly as regards the excellent methods documentation, including reproducible code.

AR1:

We are grateful to the reviewer for their in-depth assessment of the manuscript.

RC2:

My main comments concern the interpretation of the results. Presently, the discussion is rather abstract, and could have stood largely unchanged regardless what the results had been. I would invite the authors to reason explicitly about whether any other outcome could have led to a different conclusion.

AR2:

Thank you for this comment. In the discussion, we do make explicit statements about the findings and their implications for the open badges policy, for example:

“A reasonable expectation of any scientific manuscript is that repeating the original analyses upon the original data will yield the same quantitative outcomes (Bollen et al., 2015). We have found that this standard of analytic reproducibility was frequently not met in a sample of *Psychological Science* articles receiving open data badges”

“Although the open badges scheme introduced at *Psychological Science* has been associated with an increase in data availability (Kidwell et al., 2016), the current findings suggest that additional efforts may be required in order to ensure analytic reproducibility.”

It is not clear how reasoning about whether other outcomes would have led to different conclusions would be informative as this was an exercise in estimation rather than hypothesis testing.

RC3:

Personally, I think this paper contributes meaningfully towards estimating the field-wide computational reproducibility in psychology. The use of a random-effects meta-analysis implies that the authors are trying to do this, but the discussion only broaches this point indirectly, by pointing to limitations of generalizability.

AR3:

We have added some text to explicitly link our use of confidence intervals and random-effects meta-analysis to the relevant part of the discussion section:

“This study has a number of important limitations and caveats. Most pertinently, **we have reported confidence intervals and conducted a random-effects meta-analysis to aid generalization of the findings to a broader population; however, such generalizations** should be made with caution as several characteristics of our sample are likely to have had a positive impact on the reproducibility rate. We specifically selected a sample of articles for which data were already available and screened against several reusability criteria (Kidwell et al., 2016). By contrast, most psychology articles are unlikely to be accompanied by raw data or analysis scripts (Hardwicke et al., 2020a; Hardwicke & Ioannidis, 2018; Wicherts et al., 2006), and even when data is available, suboptimal management and documentation can complicate re-use (Hardwicke et al., 2018). Consequently, most psychology articles would likely fail a reproducibility check before reaching the stage of the analysis pipeline at which we began our assessments.”

RC4:

There are some other points that could also be discussed, if the authors wish. One point I find intriguing is that conclusions in the target articles often seem to be unaffected by major numerical errors. Does this imply a gap between the theoretical model and the statistical test? Is there scope for improvement here?

AR4:

This is an important point and we have addressed it in a new paragraph in the discussion as below:

Non-reproducibility does not always imply that prior conclusions based on the original results are fatally undermined – indeed we encountered no such scenarios in this study. Determining the consequences of non-reproducibility for the interpretation of the results is not always straightforward and is to some extent subjective. We considered several factors including the precision of original hypotheses, the extent and magnitude of non-reproducibility, and whether there were p-value decision errors. For example, in a case where a hypothesis only makes a directional prediction, a reanalysis indicating a statistically significant but smaller effect than originally reported, would still be consistent with the original hypothesis. Or in a case where effect sizes and p-values can be reproduced successfully, but one finds a few major numerical errors in the descriptive statistics, the most obvious cause is a reporting error, rather than an error in the implementation of the original analysis, and the original conclusions are not obviously

compromised by this. Therefore, scientific reasoning is needed to evaluate the consequences of non-reproducibility on a case-by-case basis.

RC5:

Another point of interest concerns systemic factors for quality control. The authors rightly point to the role of journals in setting policy and in conducting quality checks. They also point out resources available to scientists for generating reproducible workflows. But there is considerable scope for further discussion of systemic factors. Why, for instance, do universities send out numbers with their name on it without having had a second person double-check the work of a typically junior person? What institutional responsibilities are there? Can we (university-affiliated scientists) learn from other types of research performing organizations?

AR5:

We appreciate this comment; however, the focus of the study was assessing reproducibility in the context of a specific journal policy. We feel that a broader discussion of the many factors that potentially influence reproducibility is beyond the scope of the study.

RC6:

Finally, I wonder what the authors think that we still need to find out about computational reproducibility. Are we done with psychology now and ready to move on to other fields? Is there a research agenda that would motivate digging deeper here? For instance, would it be interesting to study covariates associated with computational reproducibility? Should we do intervention studies to evaluate effects of new policies?

AR6:

These are important questions but discussion of them is perhaps best suited to a systematic review of research on this topic. We'd prefer to keep the discussion tightly aligned to the results of the study.

RC7:

Minor points

I offer the following points at the authors' discretion.

- line 75 (about): I suggest to add here that in some cases data were missing or incorrect, and that the work led the authors of several papers to issue corrections.

AR7:

We have added the following:

In some cases, the data files contained errors or were missing values and at least one author published a correction notice.

RC8:

- line 91: I recommend to state the aim clearly here at the end of the introduction (preferably verbatim as in the preregistration: “does open data shared under the Open Sciences Badges scheme actually enable successful analytic reproducibility”).

AR8:

We have added the following:

Thus, the aim of the study was to assess the extent to which data shared under the Psychological Science open badge scheme actually enabled analytic reproducibility.

RC9:

- table 1: I suggest to use a 2x2 table instead, for better readability.

AR9:

We have converted table 1 to a 2x2 table.

RC10:

- figure 1: I suggest to draw the x axis from 0 to 1, for better readability, or else to truncate it equally at the top and bottom.

AR10:

We have converted the x axis range from 0 to 1.

RC11:

- figure 2: Why are not all p-values in upper left and lower right quadrants decision errors? I must be missing something. Can the authors please elaborate on this to clarify.

AR11:

We are grateful to the reviewer for drawing our attention to this. The value in the upper left quadrant should have been labeled as a ‘decision error’. We have corrected this.

The two values in the bottom right quadrant are marked as ‘major errors’ rather than ‘decision errors’ because of specific issues we had reproducing values from article ID 9-5-2014_PS. As

documented in vignette 25, we had difficulties reproducing the p-values in this case because the authors reported applying a multiplicity correction but did not report how it was applied (adjustment of alpha threshold or adjustment of p-values) or what the family of relevant hypothesis tests was. Because of this uncertainty, these two values are conservatively reported as major errors rather than decision errors. We have added the following to the figure caption to make this clear:

Note that the two values in the bottom right quadrant of the graph have been conservatively designated as ‘major errors’ rather than ‘decision errors’ due to uncertainty about how the original analysis was performed. Specifically, insufficient information was provided in the original article (ID 9-5-2014_PS) about how multiplicity corrections were applied (see Vignette 25 in Supplementary Information E for further information).

We have also corrected a sentence in the results section which reported “This include 1 decision error...” to the following:

This included 2 decision errors for which we obtained a statically significant p-value in contrast to a reported non-significant p-value and 1 decision error with the opposite pattern.

These changes do not impact the interpretation of the results.

RC12:

- Suppl. A: The authors write that "The most serious consequence of non-reproducibility is that it undermines the credibility of associated scientific claims." While I am inclined to agree, I do not think that this statement is obviously true. I recommend that the authors give arguments supporting this statement.

AR12:

This sentence is from the supplementary materials. The argumentation that supports it is provided in the first paragraph of the introduction.

RC13:

- line 521 (about): It would be interesting to know which of the eligibility criteria were not met in the different cases.

AR13:

As noted in the supplementary sample information, one article was not included because the data no longer appeared to be available. We have made a small adjustment to the text to make clear that the other 9 articles were excluded because

“...a finding based on a ‘relatively straightforward analysis’ could not be identified...”

RC14:

Potential conflicts of interest

I am a member of the badges committee of the Center for Open Science, which defines the criteria for earning the open science badges, and I served for several years as its chair. I was a co-author of the earlier study (Hardwicke et al. 2018) which investigated computational reproducibility of articles in the journal *Cognition* in a manner similar to this study.

AR14:

We appreciate the reviewer's transparency.

Reviewer #2

RC15:

I believe the research question the authors try to answer is a very important one in the current academic climate, in which the need for transparent research is often propagated, but rarely studied. Using badges, whether you like them or not, is one way to encourage researchers to be more open, and it's very interesting to see whether badges actually improve the quality of papers. To be clear: I think transparency is pivotal for science to enable the essential self-correcting mechanism to function properly, and I don't think that more transparency necessarily creates higher quality research, but it would be a welcome side-effect.

So I was excited to read this paper, and it does not disappoint. It is well-written, clear, thorough, and it deals well with potential shortcomings. Despite the fact that I've told a friend yesterday (before having read this paper) that I usually write long reviews, I think this one can be very short: I think the authors did a nice and painstaking job (I don't envy having to check hundreds of outcomes), and the results are useful for the current debate around transparency. So I'd have no problem with endorsing this paper, barring a few (very) minor issues:

AR15:

We are grateful to the reviewer for their assessment of the manuscript.

RC16:

- $\alpha = .05$ is taken as the criterion for decision errors. Did the authors check whether this was indeed the significance level the authors used? Usually it does, but some authors prefer other levels of α , and in that case labelling decision errors when they are on the other side of $.05$ might be premature

AR16:

We agree with the reviewer and we did take this into account – our approach was to assume the alpha boundary was .05 (as it is the de facto standard) unless otherwise stated. We have made some adjustments to make this clearer:

If an old p-value fell on the opposite side of **the alpha boundary** relative to a new p-value we additionally recorded a ‘decision error’. **The alpha boundary was assumed to be .05 unless otherwise stated.**

RC17:

- Personally, I would have put the footnote at the end of page 7 in the text: I think this is quite important

AR17:

We have moved the footnote to the main text.

RC18:

- Minor detail: the crosses in Figure 2 seem like downward diagonal lines to me, since the upward lines coincide with the main upward diagonal (so at first I thought: where are the crosses?)

AR18:

We have changed the shapes used in Figure 2 to avoid this issue.

RC19:

- Confidence intervals are weird, and confidence intervals for proportions close to 0 and 1 may be even weirder. Having a CI from $[0, X]$ makes sense given the correct interpretation of CIs (“in the long run,....”), but shouldn’t be interpreted as if 0 were a plausible value (as a matter of fact, the authors have shown it’s not 0). I have no particular suggestion how to deal with that (there is no mistake or anything), but it may be good to stress this point, possibly near Supplementary Figure 1

AR19:

Thank you, we appreciate this point; however, we also did not find any reasonable technical methods that correct for it. Facing this situation, we have added the following caveat: “Note that although this confidence interval technically includes 0, we are confident that 0 is not a possible population proportion due to the existence of non-reproducible cases that we document here.””

RC20:

But as said before, I applaud the authors for writing this paper, and it was a pleasure reading it.

I always sign my reviews.
Rink Hoekstra

AR20:

We thank the reviewer for their comments.